# Measurement and Classification Criteria of Strength Decrease Rate and Brittleness Indicator Index for Rockburst Proneness Evaluation of Hard Rocks

**DOI:** 10.3390/ma16083101

**Published:** 2023-04-14

**Authors:** Kun Du, Songge Yang, Jian Zhou, Lichang Wang

**Affiliations:** 1School of Resources and Safety Engineering, Central South University, Changsha 410083, China; 2School of Geosciences and Info-Physics, Central South University, Changsha 410083, China

**Keywords:** hard rock, rock burst tendency, lab testing, brittleness indicator, strength decrease rate

## Abstract

Rockburst is one of the common geological hazards. It is of great significance to study the evaluation indexes and classification criteria of the bursting liability of hard rocks, which is important for the prediction and prevention of rockbursts in hard rocks. In this study, the evaluation of the rockburst tendency was conducted using two indoor non-energy indexes, namely the brittleness indicator (*B*2) and the strength decrease rate (*SDR*). The measuring methods of *B* and *SDR* as well as the classification criteria were analyzed. Firstly, the most rational calculation formulas for *B* and *SDR* were selected based on previous studies. The *B*2 equaled to the ratio between the difference and sum of uniaxial compressive strength and Brazilian tensile strength of rocks. The *SDR* was the average stress decrease rate of the post-peak stage in uniaxial compression tests and equaled the uniaxial compressive strength dividing the duration time of post-peak rock failure in uniaxial compression tests. Secondly, the uniaxial compression tests of different rock types were designed and carried out, and the change trend of B and *SDR* with the increase of loading rate in uniaxial compression tests were studied in detail. The results showed that after the loading rate was greater than 5 mm/min or 100 kN/min, the *B* value was affected, limited by the loading rate, while the *SDR* value was more affected by the strain rate. The displacement control, with a loading rate of 0.1–0.7 mm/min, was recommended for the measurement of *B* and *SDR*. The classification criteria of *B*2 and *SDR* were proposed, and four grades of rockburst tendency were defined for *SDR* and *B*2 according to the test results.

## 1. Introduction

With the development of society and the economy, the needs for space and mineral resources continue to increase, and the buried depths of underground engineering or mines continue to expand [1]. Taking underground mines as an example, the shallow mineral resources have been exhausted gradually, and the deeply buried geo-resources should be mined out urgently [2,3]. The depth means high geo-stress and elastic energy stored in rocks, which makes it easier to induce rockburst. Rockburst has occurred many times in Germany, Canada, South Africa, India, Japan, and other countries [4,5,6,7,8,9]. At present, rockburst is a global underground engineering problem; it is difficult to predict accurately, and to prevent and control effectively [10]. Once rockburst occurs, it may lead to more casualties and economic losses than other disasters [2,11,12].

With in-depth study and analysis on the phenomena of rockburst, the mechanism of rockburst has been understood gradually. The occurrence of rockburst usually requires three key conditions: one is that the rock itself has high rockburst tendency; then, that the rocks are subjected to high geo-stress and possess a large amount of elastic energy; and the last is that the elastic energy stored in the rocks is induced to release by excavation, dynamic stress, or other external factors [13,14,15]. In order to accurately predict and evaluate rockburst, many scholars have introduced field tests and theoretical analysis into the research field of rockburst [16]. At present, in order to study the local strain variation and stability of rock masses, researchers have conducted rockburst mechanism studies based on capacity theory, strength theory, blast reliability analysis, and chaos theory [17,18]. However, it is difficult to establish universal and suitable criteria for rockburst prediction by simply using theoretical methods from classical mechanics. Many scholars have applied some new scientific methods to rockburst prediction, such as cloud models [19], fuzzy evaluation [20], neural networks [21], extension methods [22,23], Bayes discriminant analysis [24], and cluster analysis [25].

However, machine learning also has some defects, such as the large sample base required, unclear parameter calibration, and over-simplified models. In order to control rockburst and minimize the risk of rockburst, the evaluation of the rockburst tendency in laboratory tests has received more and more attention in recent years. The measurement methods and grade classification criteria of different bursting indexes have become the hot research field [26]. Most bursting indexes are defined from the perspective of energy, such as the strain energy storage index $W_{ET}$ [27], the peak-strength strain energy storage index $W_{ET}^{P}$, the peak-strength energy impact index $A_{CF}^{\prime }$ [28] and the residual elastic energy index $A_{EF}$ [29,30]. Likewise, there are also many non-energy bursting indexes, such as the modified brittleness index *BIM,* the deformation brittleness index $K_{u}$ [31], the decrease module index *DMI* [32], the brittleness indicator index B [33], and the strength decrease rate index SDR [34].

A multi-index bursting liability evaluation system has been formed for coal at present [35], but there is no suitable standard used for the rockburst proneness evaluation of hard rocks, especially in the aspects of the bursting indexes and classification criteria. In general, it directly adopts the standards of coal to predict the bursting proneness of hard rocks, and the prediction results are inconsistent with the field phenomena of rockburst. In fact, there are great differences between the mechanical parameters of coal and hard rocks, and the bursting proneness indexes and their criterion used in coal are not suitable for hard rocks [34]. In addition, the bursting classification criteria in hard rocks is often divided into four grades, i.e., none, weak, medium, and strong, by many researchers [26,29,30], which is a three-grade criterion for coal, i.e., none, weak, and strong. Figure 1 shows the statistics of rockburst tendency degrees of coal and non-coal rocks in underground engineering with different excavation depths in China [36,37,38,39,40,41,42,43,44,45,46,47,48,49,50,51,52,53,54,55,56,57,58,59,60,61,62]. The excavation depth of non-coal rock engineering is deeper than that of coal mines, and the intensity of rockburst in deep hard rock engineering is much stronger. Therefore, the rockburst of hard rocks should be paid more attention, accordingly.

In this study, the influence of the loading mode and rate on the *SDR*, *B*1, and *B*2 results is analyzed, and then the loading mode and rate consistent with the determination of the basic mechanical parameters of the rock are recommended. Combined with the determination results of *SDR* and *B*2 and the classification method, the rockburst tendency of different rocks is evaluated.

## 2. Definitions of *B* and *SDR*

Several non-energy indexes of rockburst tendency in laboratory tests were put forward, e.g., brittleness indicator index B, decrease module index *DMI*, deformation brittleness index $K_{u}$, and strength decrease rate index SDR [12], as shown in Table 1. The $K_{u}$ is defined as the ratio of strain corresponding to the stress at the unloading point and the terminal strain at the unloading stage, but the elastic strain at the peak is impossible to obtain, so it cannot accurately reflect the actual bursting tendency of rocks. The DMI is defined as the the ratio between pre-peak modulus and post-peak modulus, in which the pre-peak modulus reflects the rate of energy accumulation in rocks, while the post-peak modulus reflects the speed of energy release during the failure process of rocks. However, the post-peak modulus is also difficult to test and calculate. Therefore, this study focuses on the discussion of brittleness indicator index B and strength decrease rate index SDR.

The brittleness indicator is one of the important indexes for rockburst prediction, which can also be used to analyze the stability of surrounding rocks in underground engineering. The brittleness indicator index of rocks is closely related to the mineral composition, toughness, and hardness of rocks [63,64,65]. The strength decrease rate is closely related to the mechanical properties and post-failure state of rocks, can denote the release rate of elastic energy stored in rocks, and is important for the study of mechanical behaviors of rocks in the post-peak stage [66,67,68].

**Table 1 materials-16-03101-t001:** Brittleness indicator and strength decrease rate for evaluation of rockburst tendency [69].

Indicators		Formula	Parameter
SDR		$SDR=\frac{\sigma_{c}}{D_{t}}$	$\sigma_{c}$$\;\mathrm{and}\;D_{t}$ represent the uniaxial compressive strength and dynamic failure time after peak strength, respectively
*B*	B1	$B1=\frac{\sigma_{c}}{\sigma_{t}}$	$\sigma_{c}\;\mathrm{and}\;\sigma_{t}$ represent the uniaxial compressive strength and Brazilian tensile strength, respectively
	B2	$B2=\frac{\sigma_{c}-\sigma_{t}}{\sigma_{c}+\sigma_{t}}$	
$K_{u}$		$K_{u}=\frac{\varepsilon^{L}}{\varepsilon_{p}}$	$\varepsilon^{L}$$\;\mathrm{and}\;\varepsilon_{p}$ represent the strain corresponding to the stress at the unloading point and the terminal strain at the unloading stage, respectively
DMI		$DMI=\frac{E_{G}}{{|E}_{M}|}$	$E_{G}$$\;\mathrm{and}\;E_{M}$ represent the pre-peak elastic modulus and post-peak elastic modulus, respectively

The non-energy indexes of rockburst tendency were easier to be obtain and calculate than the energy indexes, e.g., strain energy storage index $W_{ET}$, energy impact index $A_{CF}$, and residual elastic energy index $A_{EF}$. This is because the peak elastic energy cannot be obtained directly through the test. The energy indexes must be obtained indirectly through the cyclic loading and unloading test, and the results of energy indexes under different unloading points have large gaps between each other. Non-energy indexes of rockburst tendency are complementary to energy indexes. The energy indexes are influenced by the energy storage and dissipation ability of rocks. The brittleness indicator index in the non-energy index is affected by the natural characteristics of rocks. The SDR is affected by the energy release rate of rocks which can impact the intensity of rockburst. Many other bursting indexes ignore the effect of energy release rate on rockburst. The non-energy index and energy index can reflect the rockburst tendency from different views, which can keep the evaluation results of rockburst tendency more comprehensively.

Different brittleness indicator indexes have different application scenarios [33]. The initial stage of rockburst is the process of crack generation and propagation. Based on Griffith strength criterion and strength parameters, the brittleness *B* of rocks was put forward to evaluate the rockburst proneness of rocks [70]. One of the calculation methods of *B* is expressed by uniaxial compressive strength and Brazilian tensile strength as Formula (1):(1)$$B1=\frac{\sigma_{c}}{\sigma_{t}}$$

The rockburst proneness criterion of *B*1 was as follows [41], which was negatively correlated with rockburst tendency.

B1>40 None;40≥B1≥26.7 Weak;26.7≥B1≥14.5 Medium;B1<14.5 Strong.

However, the other rockburst proneness criterion of *B*1 was as follows [71], which was positively correlated with rockburst tendency.

B1<15 None;15≤B1≤18 Weak;18≤B1≤22 Medium;B1>22 Strong.

The classification criteria of *B*1 were not unified, and the two categories of criteria of *B*1 disagreed with each other from the perspective of regularity [70]. Based on uniaxial compressive strength and Brazilian tensile strength, Singh [72] proposed a new method for calculating the brittleness indicator index B2, as expressed by Formula (2). Both the calculation Formulas (1) and (2) of brittleness indicator indexes B take two basic mechanical parameters, i.e., uniaxial compressive strength and Brazilian tensile strength, into account, which were affected by the fundamental mechanical properties of rocks [73,74]. There is no widely used classification criterion for *B*2 at present. Rocks with high compressive strength often have high Brazilian tensile strength, which may lead to a large dispersion of the results of *B*1, and it is difficult to judge the rockburst proneness using *B*1. According to the actual indoor rockburst situation, the rockburst tendency obtained by *B*2 is more consistent with the actual situation. Therefore, *B*2 is more suitable to use as an index of rockburst tendency.
(2)$$B2=\frac{\sigma_{c}-\sigma_{t}}{\sigma_{c}+\sigma_{t}}$$

The strength decrease rate SDR is a new index put forward by Du et al., and is defined as the ratio of uniaxial compressive strength and dynamic failure time after peak strength [22]. When the high elastic energy stored inside the rocks is induced, releasing quickly by external action, the ejection phenomenon of rock fragments is more obvious. The SDR can reflect the energy release rate of rocks to a certain extent. It is considered that among the non-energy evaluation indexes of rockburst tendency for rocks, the brittleness indicator index B and the strength decrease rate SDR can reflect the fundamental properties and energy release rate of rocks, respectively.

## 3. Loading Mode and Rate Effect on *B* and *SDR*

### 3.1. Testing Design

In order to investigate the rockburst tendency of different rocks, 4 kinds of rocks, including granite, marble, red sandstone, and coal, were selected to explore the influence of loading conditions on the brittleness indicator index B and strength decrease rate SDR. The rock specimens were prepared as cylindrical specimens, and all rock specimens were shown in Figure 2.

All specimens were drilled from intact rock blocks. The non-parallelism error of two end faces of rock specimens was not more than 0.05 mm. Along the height of the rock specimens, the diameter error was not more than 0.3 mm. The end face was perpendicular to the axis of the rock specimens, and the deviation was not greater than 0.25°. In the process of taking, transporting, and preparing rock specimens, it is necessary to avoid colliding with each other or accidentally falling to produce cracks.

Two series of testing, i.e., uniaxial compression tests and Brazilian splitting tests, were designed in the study. Before testing, rock specimens were grouped and numbered, and basic parameters, e.g., wave velocity and mass, were measured. The specimens had no defects, good uniformity and integrity, and were all in the state of natural water content. In uniaxial compression tests, the specimens were selected as cylindrical rock specimens with diameters of 50 mm, the diameters of the rock specimens were 10 times greater than the size of the largest mineral particle of rocks, and the height to diameter ratio of rock specimens was set as 2.0. In Brazil splitting test, the cylindrical rock specimens with diameter of 50 mm were used, and the ratio of height to diameter of specimens was designed as 0.5 [75].

### 3.2. Determination of SDR

The values of SDR were calculated by the results of uniaxial compression tests (Figure 3a) which were carried out on an MTS 322 T shaped material testing machine in Central South University (Figure 2); the maximum axial force applied by the testing machine was ±500 kN.

In order to avoid the loss of rock fragments and ensure security in uniaxial compression tests, a cylindrical acrylic cylinder slightly larger than the specimens was used to cover the specimens during testing. The testing machine was stopped when the specimens lost their carrying capacity.

### 3.3. Determination of B

The values of the brittleness indicator index B were calculated by the results of uniaxial compression tests and Brazilian splitting tests. The specimens in Brazil splitting tests were destroyed by radial tensile stress which was applied by a pair of steel rods with diameters less than 3 mm along the diameter of the disk specimens (Figure 3b,c). In order to make sure that the steel rods, up and down, were parallel to each other and located along the diameter with the direction of vertical loading, a novel Brazil splitting testing device was designed, machined, and applied. The detailed usage flows are shown in Figure 4.

The novel device contains four parts, i.e., V shaped steel mold, upper loading mold with steel rod, bottom loading mold with steel rod, and U-shaped outer mold with fixing screws (Figure 4a). The detailed usage flows are as follows. Firstly, the bottom loading mold with steel rod was fixed in the U-shaped outer mold. The V-shaped steel mold was used to keep the rock specimens in the middle of the U-shaped outer mold, and the fixing screws were used to fix the rock specimens (Figure 4b). Secondly, the V-shaped steel mold was removed, and the upper loading mold with steel rode was placed at the location of the V-shaped steel mold (Figure 4c). Then, the whole mold and specimen was placed on the testing machine, and the pre-loads with small values were applied on the mold (Figure 4d). At last, the fixing screws moved far away from the specimens, and test was started (Figure 4e).

### 3.4. Data Processing

According to the testing data, the stress-time curves of rock specimens (Figure 3a) were drawn, and the uniaxial compressive strength and Brazilian tensile strength of rocks were calculated (Figure 3b,c), respectively. The dynamic failure time *D*t is the time from the peak strength of rock to the time when the axial stress applied on specimens dropped to zero. According to the formulas in Table 1, the strength decrease rate *SDR* and the brittleness indicator index B of rocks were calculated directly.

### 3.5. Influence of Loading Terms on B and SDR

In order to explore the influence of loading rates on the brittleness indicator index B and strength decrease rate SDR, the displacement control mode and load control mode were selected in uniaxial compression tests. Under displacement control mode, the loading rates were pre-designed as 0.005 mm/min, 0.05 mm/min, 0.5 mm/min, 5 mm/min, 50 mm/min, and the maximum displacement loading rate of testing machine for each rock, respectively. While under load control mode, the loading rate was pre-designed as in Table 2, and each rock was divided into 6 groups in load controlled uniaxial compression tests.

In Brazilian splitting tests, the values of Brazilian tensile strength are small, and the influence of loading mode and rate on Brazilian tensile strength is ignored in this study. The load control mode with a loading rate of 0.3–0.5 MPa/s was selected in Brazilian splitting tests, which is recommended by the standard for test methods of engineering rock mass (GB/T 50266-2013) in China. The calculation results of *B* and *SDR* are shown in Table 3 and Table 4.

#### 3.5.1. Impact of Loading Rate

As shown in Figure 5a–d, with the increase of loading rate, *B*1 (calculated using Formula (1)) and B2 (calculated using Formula (2)) show a slightly increasing trend, and the values of *B*1 and *B*2 of coal and marble are larger than that of the other two rocks under load and displacement control modes.

The increase of *B*2 on granite, marble, and red sandstone is lower than 0.05, and the increase of coal rock is slightly larger (i.e., lower than 0.08). Therefore, the loading rate effects on *B*2 can be ignored. The increasements of the 4 kinds of rocks are 3–13 for *B*1, and the loading rate effects on *B*1 are clearer than on *B*2, which cannot be ignored.

The influence of loading rates on the values of the SDR is more significant, as shown in Figure 5e,f. Different types of rocks have different sensitivities to loading rate. Among the 4 kinds of rocks, the loading rate has the least influence on the SDR values of marble, which shows a linear change trend in the scope of the loading rate in this study. Under displacement control mode, the values of the SDR of granite and red sandstone increased sharply after the displacement loading rate was larger than 5 mm/min. Under load control mode, the values of the SDR of all rock types increased sharply after the loading rate was larger than 100 kN/min. The influence degrees of loading rate on the SDR values in order from the largest to the smallest are as follows: granite, marble red sandstone, and coal, which is quite different from that under displacement control mode. Thus, when the SDR is used as the evaluation index of rockburst tendency, special attention should be paid to the test within a certain range, otherwise there is no comparability of SDR obtained under different loading rates.

#### 3.5.2. Effect of Loading Mode

As shown in Figure 6, Figure 7 and Figure 8, There are certain differences between the values of *B*1, B2_,_ and *SDR* under displacement control mode and load control mode.

For *B*1 and *B*2, the loading mode effect can be ignored, and the gaps of *B*1 and *B*2 under different loading modes fluctuate to a certain extent, especially for coals. The main factor inducing the phenomena is that coal is a typical heterogeneous anisotropic sedimentary rock, and the anisotropic properties of coal specimens induce the gaps of *B*1 and *B*2 under different loading modes. Therefore, the greater the anisotropy of specimens, the more pronounced the fluctuation of *B*1 and *B*2. It is suggested that more specimens should be prepared to measure the values of *B*1 and *B*2 (Figure 6 and Figure 7).

For SDR, the control mode has great influence on the *SDR*. For granite, marble and coal, the values of the *SDR* under the load control mode are bigger than that of the *SDR* under displacement control mode. While for marble, the values of the *SDR* under displacement control mode are bigger than that of the *SDR* under loads control mode (Figure 8).

The loads applied on the specimens increase at first and then decrease, while the deformation of specimens always increases before total failure in uniaxial compression tests, so it is difficult to obtain the post-peak stage of rocks in load controlled uniaxial compression tests. When the rock specimens lose their carrying capacity after peak strength, the actuator of the testing machine under load control mode may trigger accelerated loading and shorten the duration time of the post- peak stage. The typical pre-peak and post -peak stress-time curves are shown in Figure 9.

In this study, the main aim is to determine the rockburst proneness of hard rocks by the basic mechanical parameter tests of rocks, i.e., uniaxial compression tests and Brazilian tensile tests. According to GB/T 50266-2013 “Engineering Rock Mass Test Method Standard” [75], it is recommended that the loading terms are load control mode with loading rate of 0.5–1 MPa/s, while the displacement control mode is more suitable to test the rockburst proneness index of rocks in our study. Based on the strain rate effect of rocks, the difference between the testing results of basic mechanical parameters under load control mode with loading rate of 0.5–1 MPa/s and under displacement control mode with loading rate of 0.1–0.7 mm/min is not great, and both can be used in the measurement of brittleness indicators and the strength decrease rate.

## 4. Classification Criteria of *B* and *SDR*

Twelve kinds of rocks, including white sandstone, green sandstone, grey sandstone, purple sandstone, marble, granite, Chinese black granite and brown sandstone, Andesite, yellow sandstone, red sandstone, and coal (Figure 2), were selected to determine the rockburst criterion of the brittleness indicator index *B* and the strength decrease rate *SDR*. The displacement control mode with a loading rate of 0.12 mm/min was chosen in the uniaxial compression test designed in this part, and the load control mode with a loading rate of 0.3 MPa/s was chosen in the Brazil splitting test. The obtained results of *B*1, *B*2, and *SDR* were listed in Table 5.

The rockburst tendency degrees of the rocks chosen in this study were determined by the failure phenomena. Different types of rocks will have different failure phenomena when rockburst occurs. Therefore, the classification criteria of failure phenomena are judged from two main aspects. First, the response degree of sound when the rock is about to destabilize and is damaged. The more intense or active the sound emitted near failure, the higher the rockburst tendency (an acoustic emission system is used to monitor the intensity of the sound, which is not described in detail in this manuscript due to space limitations). The second is the degree of fracture of the rock damage and the amount of debris after rock failure. The more cracks and debris produced by cracked rock, the higher the degree of rockburst tendency (debris is screened through the screen mesh). The detailed classification criterion of failure phenomena was summarized in Table 6.

In this study, the classification of rocks under different rockburst tendency indexes was based on the results of the classification criteria in Table 6. The rockburst tendency degrees of hard rocks were also divided into four grades, i.e., none, weak, medium, and strong, as shown in Figure 10. The rock samples with the same category were grouped into the same interval as far as possible, thus the dividing line under different rockburst indicators was obtained. However, the values of green sandstone calculated by the two rockburst indicators had large deviations, and its rockburst tendency evaluation was fuzzy. Therefore, we will pay more attention to and explore methods to evaluate the rockburst tendency of green sandstone in future studies.

The values of *B*1 were concentrated in the range of 5–30, and the different bursting tendency degree results of rocks based on failure phenomena criteria were cross-distributed. It was different to formulate rational classification criterion of *B*1 with a crosswise distribution trend. In this study, *B*1 was not suitable for the evaluation of rockburst tendency. According to the bursting tendency degree results of rocks based on failure phenomena criteria, the results calculated by the rockburst tendency index (i.e., *B*2 and *SDR*) are basically consistent with the experimental phenomenon classification. Therefore. the *B*2 and *SDR* can be used for the evaluation of rockburst tendency. The detailed classification criteria of *B*2 and *SDR* are as follows:

B2>0.96 None;0.96≥B2>0.93 Weak;0.93≥B2>0.89 Medium;B2≤0.89 Strong;SDR<2 MPa/s None;2 MPa/s≤SDR<6 MPa/s Weak;6 MPa/s≤SDR<20 MPa/s Medium;SDR≥20 MPa/s Strong.

## 5. Discussion 

### 5.1. Rationality of Classification Criteria

In hard rock engineering, related academic studies at home and abroad showed that rockburst often occurs in hard and brittle rocks [76]. In our study, the brittleness indicator index *B,* i.e., *B*1 and *B*2, reflects the rockburst tendency degree of rocks from aspects of the basic mechanical parameters of rocks, i.e., uniaxial compressive strength and Brazilian tensile strength. The strength decrease rate *SDR* focuses on post-peak phase of complete stress-strain curves in uniaxial compression tests, using the average stress decrease velocity to reflect the rockburst tendency degree of rocks. Compared with other non-energy indicators, these two indicators have the advantage of accurate calculation, and reflect the rockburst tendency from two different aspects. Other redundant indexes are no longer needed. In addition, the non-energy index and the energy index complement each other, and the calculation of the non-energy index is more convenient. In this study, the bursting tendency grade of white sandstone was different between the classification grade on failure phenomena criteria and that on *B*2 criterion, while the bursting tendency grades of purple sandstone and green sandstone were different between the classification grade on failure phenomena criteria and that on *SDR* criteria, as shown in Figure 10. For other rocks close to the critical value, the evaluation results are considered to be accurate. The burst tendency grade of the rocks mentioned above on *B*2 or *SDR* criteria all raised by one level, which was conducive to field rockburst protection.

### 5.2. Strain Rate Effect on Rockburst Index

It is proved that the larger the loading rate is, the bigger the rockburst index is, and the more intense the rockburst is [77,78], which is reflected by *SDR* in this study. The loading rate and mode effects can be unified by study of the strain rate effect. The strain rates of specimens under different displacement and load loading rates were discussed. As shown in Figure 11, the strain rate was positively correlated with the displacement loading rate, and the strain rate was calculated as Formula (3). The strain rates of specimens with the same height value had the same value under the same displacement loading rate.
(3)$$\varepsilon_{td}=\frac{d\varepsilon }{dt}=\frac{1}{H_{0}}\frac{dH}{dt}=\frac{D_{lr}}{H_{0}}$$
where $\varepsilon_{td}\;\mathrm{is}\;\mathrm{the}\;$strain rate of specimens under displacement control mode; *H*_0_ is the height of the specimen, 100 mm in this study; *H* is the real time height of specimens during testing; and $D_{lr}$ is the displacement loading rate.

Under loading control mode, if the same loading rate was applied for different rocks, the strain rates of different rocks were also different. The strain rate of rocks under displacement control mode showed a linear change trend (Figure 11 top), while that under load control mode had a non-linear change trend (Figure 11 bottom).

If the loading time of the pre-peak stage had a same value, the $\varepsilon_{td}$ equal the average value of the strain rate $\varepsilon_{tl}$ of specimens under load control mode. The $\varepsilon_{td}$ and $\varepsilon_{tl}$ of different rocks in uniaxial compression tests with varied loading rates are shown in Figure 12.

In this study, the *SDR* was affected greatly by the strain rate of rocks. The uniaxial compressive strength $\sigma_{c}$ increased about 1.2 times, while *Dt* increase much more than $\sigma_{c}$, as shown in Figure 13. As can be seen from the figures, the Dt starts to increase rapidly when it is larger than 0.7 mm/min. Therefore, for the displacement loading rate, 0.1–0.7 mm/min is recommended.

The classification and evaluation criteria of *B* and *SDR* in non-energy rockburst indexes are mainly discussed, while some energy-based rockburst indexes are relatively ignored. It is also found that the evaluation results of a kind of rock under different classification criteria are different. Therefore, future research will pay more attention to the comprehensive criterion of rockburst and the quantitative research of risk assessment under the aggregation of multiple indicators, such as uncertainty theory, machine learning, and numerical simulation.

## 6. Conclusions

In order to establish appropriate criteria for hard rock bursting liability, especially in terms of bursting indicators and classification criteria, the brittleness indicator index (*B*2) and strength decrease rate (*SDR*) index and their classification criteria in rockburst tendency evaluation were studied based on the results of lab testing. The research results provide reference for establishing the indoor evaluation standard of the rockburst tendency of hard rock. The conclusions of this paper are as follows:

The *B*2 and *SDR* were selected as the basic indexes for the rockburst tendency evaluation of hard rocks. It is considered that the strength decrease rate SDR reflects the speed of stress drop speed and energy release rate in rock failure stage, and the brittleness indicator index B2 value reflects the fundamental properties of rocks. The combination of the two indexes can describe rockburst tendency more comprehensively.The reasonable loading mode and rate in rockburst tendency tests were determined. After studying the influence of loading rate and loading mode on rockburst tendency, it is suggested to adopt the displacement control mode of 0.1–0.7 mm/min for rockburst tendency assessment for *B*2 and *SDR*.The classification criteria of *B*2 and *SDR* were put forward. According to the results in this study, SDR and B2 were divided into four grades. It was found that there is a strong consistency between the *B*2 index and *SDR*, which proves a rationality of the classification criteria based on the failure phenomena of rocks in uniaxial compression tests.

## Figures and Tables

**Figure 1 materials-16-03101-f001:**
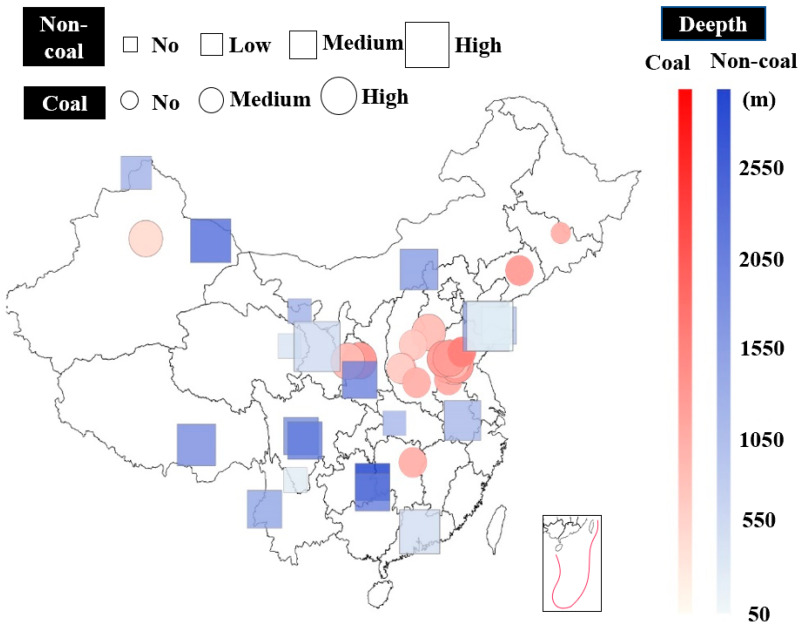
Distribution of rockburst tendency degrees of rocks in China (the darker the color, the deeper the buried depth, and the bigger the symbols, the stronger the rockburst).

**Figure 2 materials-16-03101-f002:**
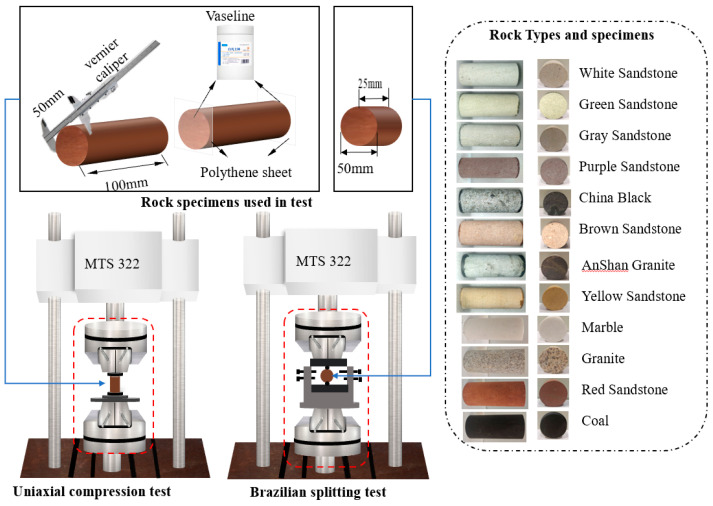
Testing machine and rock specimens used in lab test.

**Figure 3 materials-16-03101-f003:**
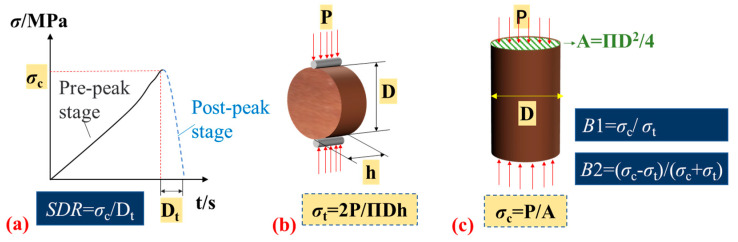
Schematic diagram of the SDR and B calculation process: (**a**) stress-time curve of rocks in uniaxial compression test and determination of *SDR*, (**b**) determination of *σ*_t_; (**c**) determination of *σ*_c_, *B*1, and *B*2. SDR denotes the strength decrease rate (MPa/s); $D_{t}$ denotes the dynamic failure time in uniaxial compression test (s); $\sigma_{c}$ denotes the uniaxial compressive strength of rock specimens (MPa); $\sigma_{t}$ denotes the Brazilian tensile strength of rock specimens (MPa); P denotes the peak load in uniaxial compression test (kN); A denotes the area of the bearing section of a rock specimen (mm^2^); D denotes the cross sectional diameter of a rock specimen (mm); h denotes the height of the rock specimen (mm).

**Figure 4 materials-16-03101-f004:**
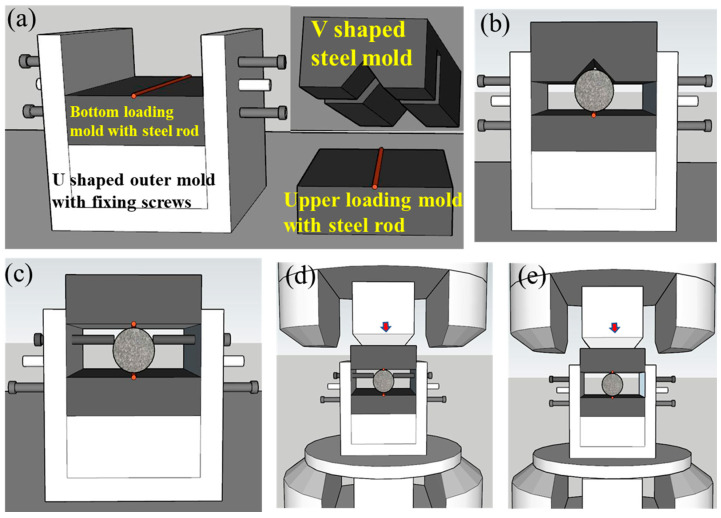
Composition and usage flow of a novel Brazil splitting testing device: (**a**) U Shaped outer steel mold with fixing screws, (**b**) V Shaped steel mold and upper loading mold with steel rod; (**c**) Install rock specimen, (**c**) Change V shaped steel mold as upper loading mode, (**d**) Install the entire mold, (**e**) Withdraw fixing screws and loading.

**Figure 5 materials-16-03101-f005:**
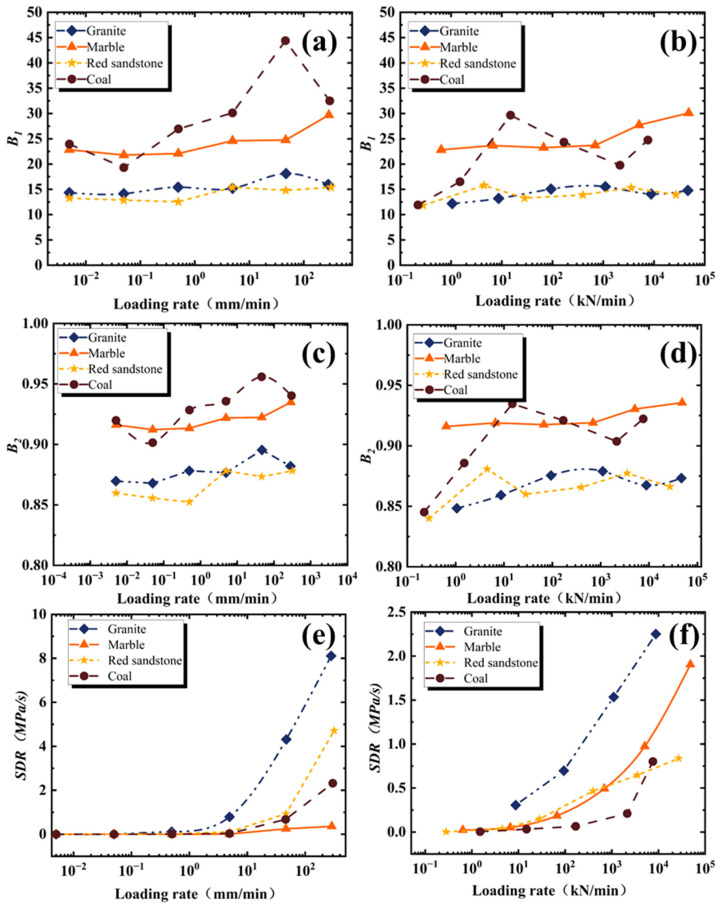
Effect of loading rate on B1, B2, and *SDR*: (**a**) *B*1 under displacement control mold, (**b**) *B*1 under load control mold, (**c**) *B*2 under displacement control mold, (**d**) *B*2 under load control mold, (**e**) *SDR* under displacement control mold, (**f**) *SDR* under load control mold.

**Figure 6 materials-16-03101-f006:**
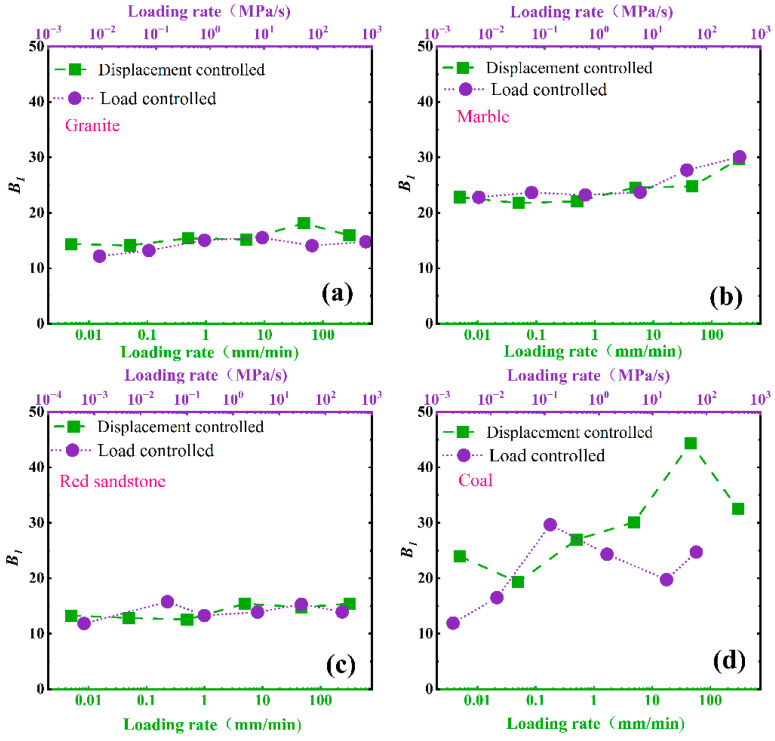
Effect of loading mode on *B*1: (**a**) granite, (**b**) marble, (**c**) red sandstone, (**d**) coal.

**Figure 7 materials-16-03101-f007:**
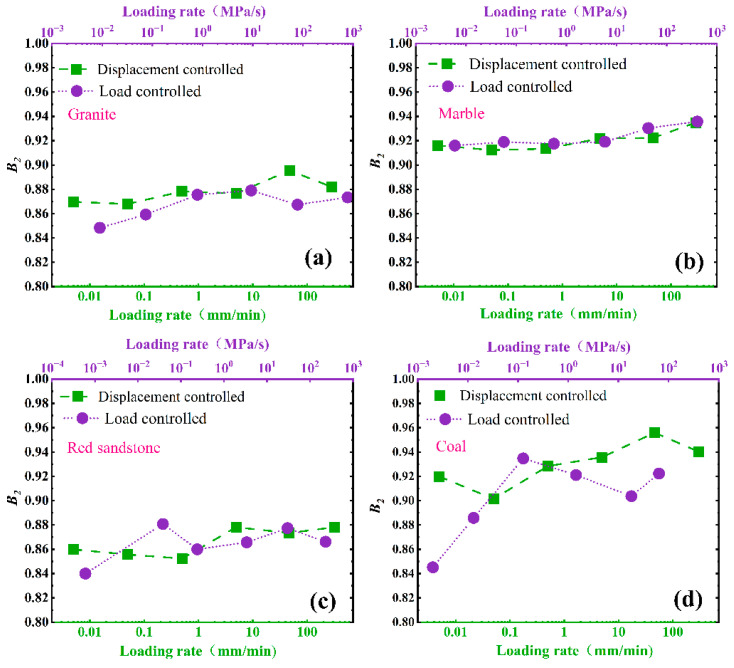
Effect of loading mode on *B*2: (**a**) granite, (**b**) marble, (**c**) red sandstone, (**d**) coal.

**Figure 8 materials-16-03101-f008:**
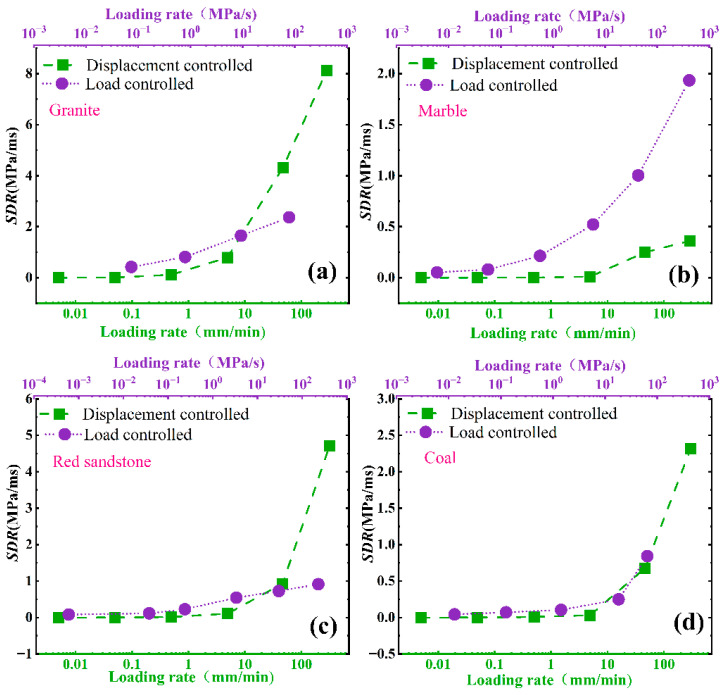
Effect of loading mode on SDR: (**a**) granite, (**b**) marble, (**c**) red sandstone, (**d**) coal.

**Figure 9 materials-16-03101-f009:**
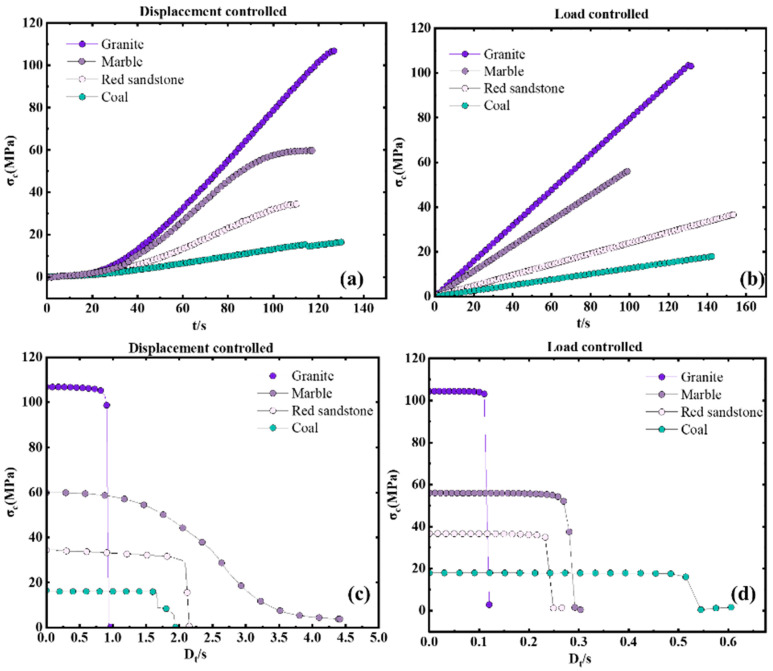
Stress-time curves of rocks in uniaxial compression tests (**a**,**c**) with a loading rate of 0.5 mm/min; (**b**,**d**) with a loading rate of 93 kN/min.

**Figure 10 materials-16-03101-f010:**
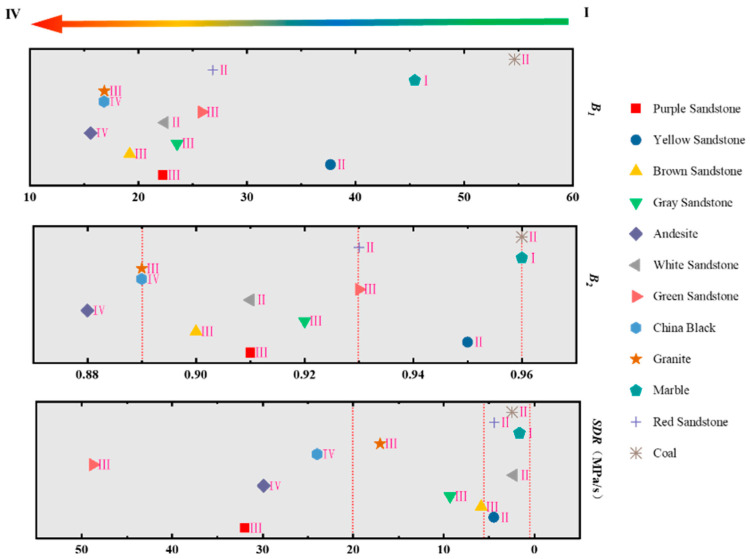
Classification criteria of the rockburst proneness of rocks in this study.

**Figure 11 materials-16-03101-f011:**
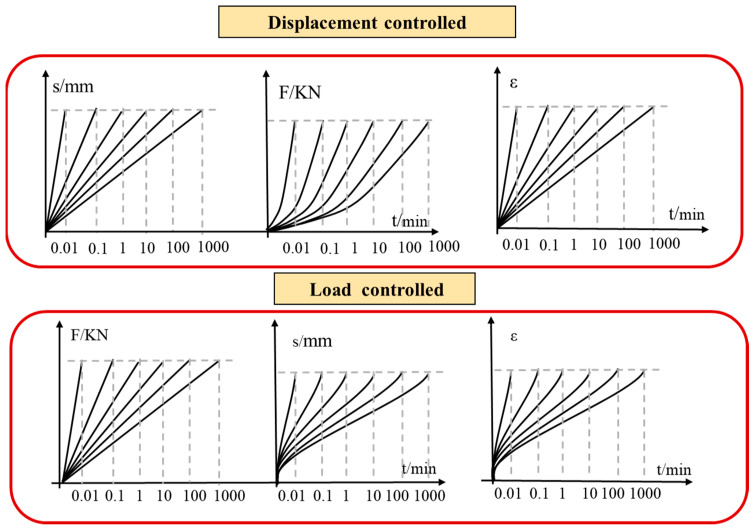
Relationship between strain rate under displacement controlled and load control mode.

**Figure 12 materials-16-03101-f012:**
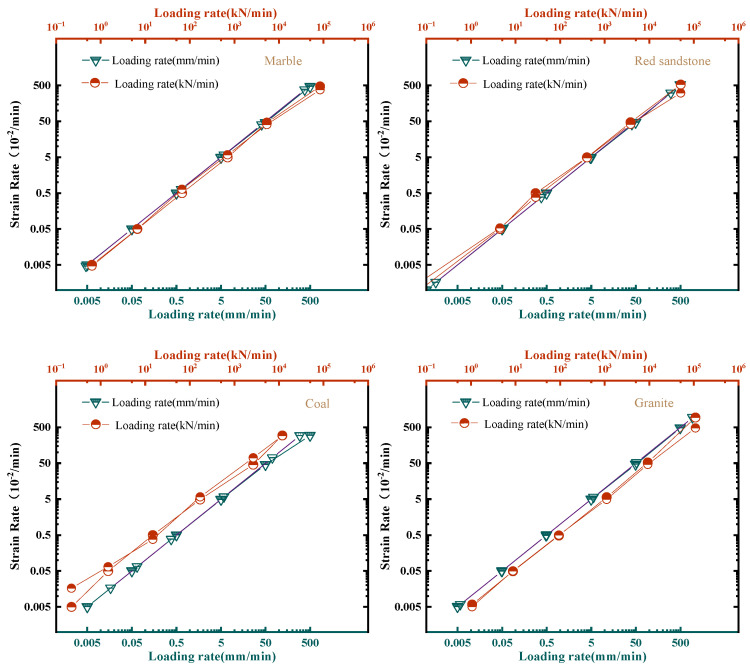
Relationship between experimental rate and strain rate of different rocks.

**Figure 13 materials-16-03101-f013:**
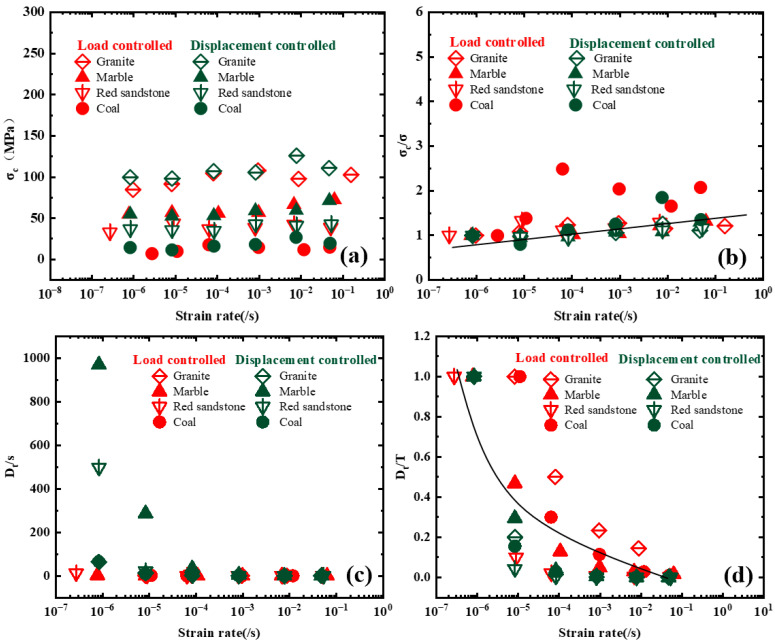
Relationships between (**a**) uniaxial compressive strength vs. time, (**b**) normalization uniaxial compressive strength vs. time, (**c**) Dt vs. time, and (**d**) normalization Dt vs. time.

**Table 2 materials-16-03101-t002:** Loading rate in load controlled uniaxial compression tests.

**Rock** **Lithology**	**Loading Rate of Each Group**								
	**Group 1**			**Group 2**			**Group 3**		
	**kN/min**	**MPa/s**	**Sample** **Number**	**kN/min**	**MPa/s**	**Sample** **Number**	**kN/min**	**MPa/s**	**Sample** **Number**
Granite	1.06	0.009	1	8.72	0.074	1	93.48	0.7935	1
Marble	0.64	0.006	1	6.66	0.057	1	66.67	0.5659	1
Red sandstone	0.07	0.0006	1	4.49	0.038	1	28.17	0.2391	1
Coal	0.22	0.002	1	1.50	0.013	1	14.87	0.1262	1
**Rock** **Lithology**	**Group 4**			**Group 5**			**Group 6**		
	**kN/min**	**MPa/s**		**kN/min**	**MPa/s**		**kN/min**	**MPa/s**	
Granite	1101.60	9.3507	1	8799.93	78.383	1	46,683.47	778.0578	1
Marble	693.32	5.8857	1	5082.66	43.1430	1	48,380.29	410.6647	1
Red sandstone	397.46	3.37377	1	3540.87	30.0558	1	27,172.56	230.6478	1
Coal	168.14	1.4273	1	2135.74	18.1287	1	7590.34	64.4288	1

**Table 3 materials-16-03101-t003:** Calculation results of *B*1, *B*2, and *SDR* under different loading rates in displacement controlled uniaxial compression tests.

Rock Lithology	Loading Rate (mm/min)	B1 Formula (1)	B2 Formula (2)	*SDR* (MPa/s)
Granite	0.005	14.35	0.87	1.57
Granite	0.05	14.14	0.87	7.72
Granite	0.49	15.42	0.88	114.25
Granite	4.94	15.20	0.88	795.80
Granite	47.18	18.12	0.90	4299.82
Granite	281.23	15.95	0.88	8109.59
Marble	0.005	22.83	0.92	0.06
Marble	0.05	21.78	0.91	0.18
Marble	0.50	22.08	0.91	1.61
Marble	4.93	24.59	0.92	11.08
Marble	46.52	24.75	0.92	250.26
Marble	288.25	29.69	0.93	362.04
Red sandstone	0.001	13.27	0.86	0.07
Red sandstone	0.05	12.86	0.86	1.69
Red sandstone	0.50	12.54	0.85	16.25
Red sandstone	4.94	15.41	0.88	108.82
Red sandstone	46.63	14.80	0.87	932.70
Red sandstone	316.67	15.41	0.88	4627.39
Coal	0.005	23.95	0.92	0.22
Coal	0.05	19.30	0.90	1.16
Coal	0.50	26.95	0.93	8.47
Coal	4.93	30.12	0.94	32.00
Coal	46.04	44.38	0.96	676.21
Coal	298.95	32.50	0.94	2320.13

**Table 4 materials-16-03101-t004:** Calculation results of *B*1, *B*2, and *SDR* under different loading rates in load controlled uniaxial compression tests.

Rock Type	Loading Rate (kN/min)	B1 Formula (1)	B2 Formula (2)	*SDR* (MPa/s)
Granite	1.06	12.19	0.85	/
Granite	8.72	13.20	0.86	305.62
Granite	93.48	15.06	0.88	697.20
Granite	1101.60	15.53	0.88	1534.98
Granite	8799.93	14.08	0.87	2251.52
Granite	46,683.47	14.78	0.87	--
Marble	0.64	22.81	0.92	23.38
Marble	6.66	23.66	0.92	51.82
Marble	66.67	23.24	0.92	184.78
Marble	693.32	23.71	0.92	492.96
Marble	5082.66	27.70	0.93	973.80
Marble	48,380.29	30.08	0.94	1904.80
Red sandstone	0.28	11.84	0.84	2.75
Red sandstone	4.49	15.77	0.88	36.57
Red sandstone	28.17	13.28	0.86	148.54
Red sandstone	397.46	13.89	0.87	466.09
Red sandstone	3540.87	15.31	0.88	646.69
Red sandstone	27,172.56	13.94	0.87	837.16
Coal	0.22	11.91	0.85	--
Coal	1.50	16.51	0.89	5.54
Coal	14.87	29.66	0.93	33.20
Coal	168.14	24.33	0.92	71.36
Coal	2135.74	19.77	0.90	230.81
Coal	7590.34	24.74	0.92	835.25

**Table 5 materials-16-03101-t005:** Calculation results of *B*1, *B*2, and *SDR*.

Rock Lithology	Sample Number	Brazilian Tensile Strength (MPa)	Uniaxial Compressive Strength (MPa)	B1	B2	SDR (MPa/s)
Purple sandstone	1	4.27	95.04	22.26	0.91	31.98
Yellow sandstone	1	1.16	43.72	37.69	0.95	4.48
Brown sandstone	1	3.38	64.92	19.21	0.90	5.86
Grey sandstone	1	2.99	70.45	23.56	0.92	9.3
Andesite	1	12.85	200.59	15.61	0.88	29.91
White sandstone	1	1.64	36.80	22.44	0.91	2.29
Green sandstone	1	2.60	67.15	25.83	0.93	48.77
Chinese black granite	1	8.97	150.98	16.83	0.89	24
Granite	1	6.68	112.65	16.86	0.89	17.03
Marble	1	2.41	109.54	45.45	0.96	1.63
Red sandstone	1	2.79	74.90	26.85	0.93	4.41
Coal	1	0.61	33.33	54.64	0.96	2.45

**Table 6 materials-16-03101-t006:** Rockburst classification criterion based on the failure phenomena of hard rocks.

Category	I(None)	II(Weak)	III(Medium)	IV(Strong)
Phenomenon	One or two penetrating cracks appeared on the specimen surface after failure. Few fragments formed with slight sound.	Multiple cracks appeared on the specimen surface after failure. A small number of fragments dropped slowly with a slight sound.	Multiple cracks appeared on the specimen surface after failure. Several fragments ejected with a clear failure sound.	Multiple cracks appeared on the specimen surface after failure. A large number of fragments ejected with great failure sound.
Classification results	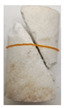 Marble	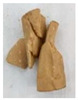 Yellow sandstoneOther rocks:Red sandstone, White sandstone,Coal	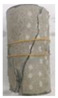 Grey sandstone Other rocks:Brown sandstone, Green sandstonePurple sandstoneGranite	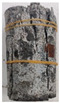 Chinese black graniteOther rocks:Andesite

## Data Availability

The data that support the findings of this study are available on request from the corresponding author, [wlccsu@csu.edu.cn], upon reasonable request.

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
