# Peer review of "Measurement and Classification Criteria of Strength Decrease Rate and Brittleness Indicator Index for Rockburst Proneness Evaluation of Hard Rocks"

_materials, 2023, doi:10.3390/ma16083101_

Round 1

Reviewer 1 Report (Previous Reviewer 2)

Authors made extensive corrections of the first version of the Ms. The Ms can be accepted for the publication

Author Response

to Reviewer’ Comments

Reviewer 1Comments and Suggestions for Authors

Authors made extensive corrections of the first version of the Ms. The Ms can be accepted for the publication

Reply: Thanks very much for your comments. Thank you again for your approval of our revision.

Reviewer 2 Report (New Reviewer)

In this manuscript, the criteria of strength decrease rate (SDR) and brittleness indicator (B) index for rockburst proneness evaluation of hard rocks are tested and the influence of the loading rate and loading mode on the SDR and B are analyzed. Although this manuscript lacks novelty, it is partly interesting for readers. I recommend that this manuscript should have a substantial revision. My comments are as follows:

1.      In the abstract, the definition for the SDR and B is the previous results, not proposed by the authors.

2.      The authors should list the specimen number for each group in Table 2 to avoid the discreteness of test results.

3.      Are the results in Tables 3 and 4 average? If they are average, the authors should list the standard deviation.

4.      The curve in Fig. 5 should be a point line graph, rather than a fitting curve.

5.      In line 247, the increment of the coal for B2 is larger than 0.05 from Fig. 5(d).

6.      The format of Fig. 8 should be the same as other figures.

7.      The unit in Figs. 6 and 7 was missed. The unit for the loading rate should be MPa/s or MPa/min. In addition, the format of ticks in Figs. 6-8 should be notated by scientific.

8.      The authors should list the specimen number for each group and the standard deviation of the results in Table 5. The authors also need to explain why the values of B1 in Tables 2 and 5  are much different for marble, red sandstone, and coal.

9.      The rock failure phenomena are different when rockburst occurs. For Table 6, the authors need to explain the reason and mechanism for the category.

10.  In line 328, it should be B2. The category for each type rock can be deleted, then plot the category region based on the values of B2 and SDR in Fig. 10.

11.  With different indicators, the classification results may be different, such as Green Sandstone. Then how to determine the category?

Author Response

to Reviewer’ Comments

Reviewer 2Comments and Suggestions for Authors

In this manuscript, the criteria of strength decrease rate (SDR) and brittleness indicator (B) index for rockburst proneness evaluation of hard rocks are tested and the influence of the loading rate and loading mode on the SDR and B are analyzed. Although this manuscript lacks novelty, it is partly interesting for readers. I recommend that this manuscript should have a substantial revision. My comments are as follows:

1. In the abstract, the definition for the SDR and B is the previous results, not proposed by the authors.

Reply: Thanks very much for your comments.

Right, "Firstly, the most rational calculation formulas for B and SDR were defined" was replaced by "Firstly, the most rational calculation formulas for B and SDR were selected based on previous studies."

2. The authors should list the specimen number for each group in Table 2 to avoid the discreteness of test results.

Reply: Thanks very much for your comments.

Right, because many different loading rates were designed in the experiment of this manuscript, only one sample was tested in each group. The sample numbers of each group have been added to Table 2.

3. Are the results in Tables 3 and 4 average? If they are average, the authors should list the standard deviation.

Reply: Thanks very much for your comments.

Because only one sample was tested at each loading rate, the results given in Tables 3 and 4 are exact values and not average values.

4. The curve in Fig. 5 should be a point line graph, rather than a fitting curve.

Reply: Thanks very much for your comments.

Right, Figure 5 has been modified as a point line graph as suggested.

5. In line 247, the increment of the coal for B2 is larger than 0.05 from Fig. 5(d).

Reply: Thanks very much for your comments.

Right, the coal rock (soft rock) had a maximum increment of 0.08 compared to the other three rocks (hard rock). Therefore, line 247 is rewritten as:” The increase of B2 on granite, marble and red sandstone is lower than 0.05, and the increase of coal rock is slightly larger (i.e., lower than 0.08). Therefore, the loading rate effects on B2 can be ignored.”

6. The format of Fig. 8 should be the same as other figures.

Reply: Thanks very much for your comments.

Right, the format of Figure 8 has been modified according to your requirements to make it consistent with the format of Figure 6 and 7.

7. The unit in Figs. 6 and 7 was missed. The unit for the loading rate should be MPa/s or MPa/min. In addition, the format of ticks in Figs. 6-8 should be notated by scientific.

Reply: Thanks very much for your comments.

Right, the missing units in Figures 6 and 7 were re-added, and the unit of force-controlled loading rate was replaced with MPa/s. In addition, the scale format in Figure 6-8 is scientifically annotated.

8. The authors should list the specimen number for each group and the standard deviation of the results in Table 5. The authors also need to explain why the values of B1 in Tables 2 and 5 are much different for marble, red sandstone, and coal.

Reply: Thanks very much for your comments.

Right, in this section of the tests, only one sample of each rock was used and the results obtained have no standard deviation. The number of samples of each sample has been added to Table 5. The 12 rocks in Section 4 were tested in a displacement control mode with a loading rate of 0.12 mm/min in the uniaxial compression test and with a loading control mode with a loading rate of 0.3 MPa/s in the Brazilian splitting test, which is not the same batch as the samples tested in Table 2. Therefore, there are differences in the results obtained. The relevant text is described in lines 305-311.

9. The rock failure phenomena are different when rockburst occurs. For Table 6, the authors need to explain the reason and mechanism for the category.

Reply: Thanks very much for your comments.

Right, different types of rocks will have different failure phenomena when rockburst occur. Therefore, the classification criteria of failure phenomena are judged from two main aspects: first, the response degree of sound when the rock is about to destabilize and damaged. The more intense or active the sound emitted near failure, the higher the rockburst tendency (Acoustic emission system is used to monitor the intensity of the sound, which is not described in detail in this manuscript due to space limitations). The second is the degree of fracture of the rock damage and the number of debris after rock failure. The more cracks and debris produced by cracked rock, the higher the degree of rockburst tendency (debris are screened through the screen mesh). The above description is added to lines 316-324.

10. In line 328, it should be B2. The category for each type rock can be deleted, then plot the category region based on the values of B2 and SDR in Fig. 10.

Reply: Thanks very much for your comments.

Right, the "B1" in line 336 (originally 328) is replaced by "B2". It is not recommended to remove the category for each rock, as it is needed to identify whether the three rockburst indicators are suitable for the evaluation of rockburst tendency. If the category for each rock is deleted, it is difficult to find the conclusion that "the B1 was not suitable for the evaluation of rockburst tendency". This manuscript not only establishes the classification criteria for rockburst by the failure phenomena of the rocks, but also gives the category areas in lines 347-354 based on the values of B2 and SDR.

11. With different indicators, the classification results may be different, such as Green Sandstone. Then how to determine the category?

Reply: Thanks very much for your comments.

Right, in this manuscript, the classification of rocks under different rockburst tendency indexes is based on the results of the classification criteria in Table 6. The 12 rocks are arranged in Figure 10 by the values of rockburst indicators, and the rock samples with the same category are grouped into the same interval as far as possible, thus the dividing line under different rockburst indicators is obtained. However, the values of green sandstone calculated by the two rockburst indicators have large deviations, and its rockburst tendency evaluation is fuzzy. Therefore, we will pay more attention to and explore methods to evaluate the rockburst tendency of green sandstone in the future studies. The above description is added to lines 327-335. Some other rocks with taxonomic differences are discussed in Section 5.1 lines 367-374.

Reviewer 3 Report (New Reviewer)

Dear Authors,

the article is relevant and, in my opinion, quite interesting for the specialists in the field. However, there are some notes that, I sippuse, should be revised before next promotion:

1. Common description of the methodology should be given, to show substantiation of the method used for the readers.

2. There is no methodological result in the proper section, so the adequatecy of the method used is not proven.

3. In my opinion, the Discussion section in your manuscript is a continuation of Results section.  However, Discussion section should contain your advances over existing studies, the limitations od reserach and its future horizon.

4. References list should contain, in my opinion, more international and new sources. 

Good luck!

Author Response

to Reviewer’ Comments

Reviewer 3Comments and Suggestions for Authors

Dear Authors,

the article is relevant and, in my opinion, quite interesting for the specialists in the field. However, there are some notes that, I sippuse, should be revised before next promotion:

1. Common description of the methodology should be given, to show substantiation of the method used for the readers.

Reply: Thanks very much for your comments.

The influence of the loading mode and rate on the SDR and B2 results is analyzed, and then the loading mode and rate consistent with the determination of the basic mechanical parameters of the rock are recommended. Combined with the determination results of SDR and B2 and the classification method, the rockburst tendency of different rocks is evaluated.

The relevant text description was added to lines 73-76.

2. There is no methodological result in the proper section, so the adequatecy of the method used is not proven.

Reply: Thanks very much for your comments.

The classification of rock burst tendency of 13 typical rocks has been evaluated, and the results are in agreement with the classification of experimental phenomena. It indicates that the classification methods of SDR and B2 for rockburst tendency formulated in this paper are applicable. The relevant text description was added to lines 342-346.

3. In my opinion, the Discussion section in your manuscript is a continuation of Results section. However, Discussion section should contain your advances over existing studies, the limitations od reserach and its future horizon.

Reply: Thanks very much for your comments.

Right, The Discussion section of the manuscript is a continuation of the Results section, but also a further in-depth analysis of the research results. In this section, we determine the results of rockburst evaluation for some fuzzy graded rock samples, and the suggested displacement loading rates are obtained from the analysis.

In terms of limitations and future prospects of the study, we added the paragraph " The classification and evaluation criteria of B and SDR in non-energy rockburst indexes are mainly discussed, while some energy-based rockburst indexes are relatively ignored. It is also found that the evaluation results of a kind of rock under different classification criteria are different. Therefore, future research will pay more attention to the comprehensive criterion of rockburst and the quantitative research of risk assessment under the aggregation of multiple indicators, such as uncertainty theory, machine learning and numerical simulation." in lines 404-410.

References list should contain, in my opinion, more international and new sources. Good luck!

Reply: Thanks very much for your comments.

Right, we have removed some of the old references and accordingly added some papers from recent years. The specific list of references is as follows:

[8] D. Malan, J. Napier, A limit equilibrium fracture zone model to investigate seismicity in coal mines, Int J Min Sci Techno, 28 (2018) 745-753.

[12] A. Voza, L. Valguarnera, R. Marrazzo, G. Ascari, D. Boldini, A New In Situ Test for the Assessment of the Rock-Burst Alarm Threshold During Tunnelling, Rock Mech Rock Eng, 56 (2022) 1645-1661.

[14] S.-f. Wang, L.-c. Sun, Y. Tang, Y. Jing, X.-b. Li, J.-r. Yao, Field application of non-blasting mechanized mining using high-frequency impact hammer in deep hard rock mine, Transactions of Nonferrous Metals Society of China, 32 (2022) 3051-3064.

[15] S. Wang, X. Li, J. Yao, F. Gong, X. Li, K. Du, M. Tao, L. Huang, S. Du, Experimental investigation of rock breakage by a conical pick and its application to non-explosive mechanized mining in deep hard rock, Int J Rock Mech Min, 122 (2019).

[16] P. X. Li, B. R, Chen, Y. Y. Zhou,Y. Xiao,G. L. Feng, G. Zhu, Research Progress on prediction and warning of hard rock burst, Journal of China Coal Society, 44 (2019) 19.

[17] F. Gong, S. Luo, Q. Jiang, L. Xu, Theoretical verification of the rationality of strain energy storage index as rockburst criterion based on linear energy storage law, J Rock Mech Geotech, 14 (2022) 1737-1746.

[18] S. Liu, F. Xiao, T. Li, B. Zhang, Analysis of Impact Tendency and Sensitivity of Fractured Rock with Different Crack Arrest Measures, Sustainability, 14 (2022).

[21] Q. Kang, Y. Xia, M. Shi, W. Zhang, W. Wang, D. Kong, Y. Wang, D.E. Manolakos, Evaluation of Rock Burst Propensity and Rock Burst Mechanism in Deep Phosphate Mines: A Case Study of Sujiapo Phosphate Mine, Hubei Province, China, Adv Mater Sci Eng, 2022 (2022) 1-13.

[22] W.Z.Liang,G. Y. Zhao, Long and short term rockburst risk assessment of deep hard rock, Chinese Journal of Rock Mechanics and Engineering, 41 (2022) 21.

[25] Methods for determination, monitoring and control of rock burst - Part 2: Classification and index determination of coal bursting liability, 2010.

[26] Z. G. Cui, Rock burst tendency evaluation and analysis based on laboratory test, Northeast Water Resources and Hydropower, 37 (2019) 3.

[59] L.N.Y. Wong, F. Meng, T. Guo, X. Shi, The Role of Load Control Modes in Determination of Mechanical Properties of Granite, Rock Mech Rock Eng, 53 (2019) 539-552.

Reviewer 4 Report (New Reviewer)

The following comments need to be incorporated for further strengthen of research paper

1. First sentence of abstract need to be rephrased for easy understanding of readers 

2. 2nd sentence of the abstract must be decided into two sentences

3. Abstract need to be modified mentioning the main findings of the research

4. Author's are advised to mention in the introduction section that why they have selected the two method as number of methods and indexes are available in the literature for measuring of rock burst.

5. Introduction section need to be updated by including latest literature as number of research papers available on rock burst 

6. Figure 2 contents need to be labelled

7. Mention significance of research in conclusion 

Author Response

to Reviewer’ Comments

Reviewer 4Comments and Suggestions for Authors

The following comments need to be incorporated for further strengthen of research paper

1. First sentence of abstract need to be rephrased for easy understanding of readers

Reply: Thanks very much for your comments.

Right, the original manuscript has been rewritten that is " Rock burst is one of the common geological hazards. It is of great significance to study the evaluation indexes and classification criteria of bursting liability of hard rocks, which is important for the prediction and prevention of rock bursts in hard rocks.” in lines 9-11.

2. 2nd sentence of the abstract must be decided into two sentences

Reply: Thanks very much for your comments.

Right, the original manuscript has been rewritten that is “In this study, the evaluation of rockburst tendency was conducted using two indoor non-energy indexes, namely brittleness indicator (B) and strength decrease rate (SDR). The methods of measuring B and SDR as well as their classification criteria were analyzed.” in lines 11-13.

3. Abstract need to be modified mentioning the main findings of the research

Reply: Thanks very much for your comments.

Right, the original manuscript has been rewritten that is “The results showed that: After the loading rate was greater than 5 mm/min or 100 kN/min, the B value was limited by the loading rate, while the SDR value was more affected by the strain rate. 0.1-0.7 mm/min displacement control was recommended for the measurement of B and SDR. The classification criteria of B2 and SDR were proposed, and four grades of rockburst tendency were defined by SDR and B2 according to the test results.” in lines 20-25.

4. Author's are advised to mention in the introduction section that why they have selected the two method as number of methods and indexes are available in the literature for measuring of rock burst.

Reply: Thanks very much for your comments.

In this manuscript, the focus is on the non-energy impact indexes of rockburst tendency. Among these non-energy impact indicators, B and SDR were selected to evaluate the rockburst tendency, and the reasons for selection are described in detail in lines 77-88 of the original manuscript. Therefore, it is not necessary to repeat the description in the introduction.

5. Introduction section need to be updated by including latest literature as number of research papers available on rock burst

Reply: Thanks very much for your comments.

Right, we have removed some of the old references and accordingly added some papers from recent years. The specific list of references is as follows:

[8] D. Malan, J. Napier, A limit equilibrium fracture zone model to investigate seismicity in coal mines, Int J Min Sci Techno, 28 (2018) 745-753.

[12] A. Voza, L. Valguarnera, R. Marrazzo, G. Ascari, D. Boldini, A New In Situ Test for the Assessment of the Rock-Burst Alarm Threshold During Tunnelling, Rock Mech Rock Eng, 56 (2022) 1645-1661.

[14] S.-f. Wang, L.-c. Sun, Y. Tang, Y. Jing, X.-b. Li, J.-r. Yao, Field application of non-blasting mechanized mining using high-frequency impact hammer in deep hard rock mine, Transactions of Nonferrous Metals Society of China, 32 (2022) 3051-3064.

[15] S. Wang, X. Li, J. Yao, F. Gong, X. Li, K. Du, M. Tao, L. Huang, S. Du, Experimental investigation of rock breakage by a conical pick and its application to non-explosive mechanized mining in deep hard rock, Int J Rock Mech Min, 122 (2019).

[16] P. X. Li, B. R, Chen, Y. Y. Zhou,Y. Xiao,G. L. Feng, G. Zhu, Research Progress on prediction and warning of hard rock burst, Journal of China Coal Society, 44 (2019) 19.

[17] F. Gong, S. Luo, Q. Jiang, L. Xu, Theoretical verification of the rationality of strain energy storage index as rockburst criterion based on linear energy storage law, J Rock Mech Geotech, 14 (2022) 1737-1746.

[18] S. Liu, F. Xiao, T. Li, B. Zhang, Analysis of Impact Tendency and Sensitivity of Fractured Rock with Different Crack Arrest Measures, Sustainability, 14 (2022).

[21] Q. Kang, Y. Xia, M. Shi, W. Zhang, W. Wang, D. Kong, Y. Wang, D.E. Manolakos, Evaluation of Rock Burst Propensity and Rock Burst Mechanism in Deep Phosphate Mines: A Case Study of Sujiapo Phosphate Mine, Hubei Province, China, Adv Mater Sci Eng, 2022 (2022) 1-13.

[22] W.Z.Liang,G. Y. Zhao, Long and short term rockburst risk assessment of deep hard rock, Chinese Journal of Rock Mechanics and Engineering, 41 (2022) 21.

[25] Methods for determination, monitoring and control of rock burst - Part 2: Classification and index determination of coal bursting liability, 2010.

[26] Z. G. Cui, Rock burst tendency evaluation and analysis based on laboratory test, Northeast Water Resources and Hydropower, 37 (2019) 3.

[59] L.N.Y. Wong, F. Meng, T. Guo, X. Shi, The Role of Load Control Modes in Determination of Mechanical Properties of Granite, Rock Mech Rock Eng, 53 (2019) 539-552.

6.Figure 2 contents need to be labelled

Reply: Thanks very much for your comments.

Right, the contents in Figure 2 contents have been labelled

7. Mention significance of research in conclusion

Reply: Thanks very much for your comments.

Right, the original manuscript has been rewritten that is “in order to establish appropriate criteria for hard rock bursting liability, especially in terms of bursting indicators and classification criteria” in 415-416.

Round 2

Reviewer 2 Report (New Reviewer)

There is only one specimen. The calculation results in Table. 5  are not average value.

Author Response

to Reviewer’ Comments

Reviewer 3There is only one specimen. The calculation results in Table. 5  are not average value.

Reply: Thanks very much for your comments.

       Right, in this section of the tests, only one sample of each rock was used and the results obtained have no average value. Therefore, line 323 is rewritten as: “The obtained results of B1, B2, and SDR were listed in Table 5.. ”

Reviewer 3 Report (New Reviewer)

Dear Authors,

the paper was significantly improved, however, I suggest adding some more international references.

Good luck!

Author Response

to Reviewer’ Comments

Reviewer 3the paper was significantly improved, however, I suggest adding some more international references.

Reply: Thanks very much for your comments.

       According to your suggestion, we have added some international references to the manuscript. The relevant contents are added to lines 44-57, and the list of added literature is as follows:

[16] Y. Sun, B. Tian, X. Liu, D. Chen, Y. Wu, A Kind of Prediction Based on SOM Neural Clustering and Combination Weighting Evaluation Method. Companion Proceedings of the Web Conference 2021. Ljubljana Slovenia: ACM, 2021: 90–96.

[17] J. Zhou, X. Li, X. Shi, Long-term prediction model of rockburst in underground openings using heuristic algorithms and support vector machines, Safety science, 2012, 50(4): 629–644.

[18] C. Tang, J. Wang, J. Zhang, Preliminary engineering application of microseismic monitoring technique to rockburst prediction in tunneling of Jinping II project, J Rock Mech Geotech, 2010, 2(3): 193–208.

[19] H. Liu, Classified prediction model of rockburst using rough sets-normal cloud, Neural computing & applications, 2019, 31(12): 8185–8193.

[20] W. Liang, G. Zhao, W. Hao, D. Bing, Risk assessment of rockburst via an extended MABAC method under fuzzy environment, Tunn Undergr Sp Tech, 83 (2019) 533-544.

[21] Z. Xuan, X. Bu, The Forecasting of Rockburst in Deep-buried Tunnel with Adaptive Neural Network,  International Conference on Industrial and Information Systems. Haikou, China: IEEE, 2009: 3–6.

[22] J. Chen, Y. Chen, S. Yang, X. Zhong, X. Han, A prediction model on rockburst intensity grade based on variable weight and matter-element extension, PLoS ONE, 2019, 14(6): e0218525.

[23] L. Zhang, X. Zhang, J. Wu, D. Zhao, H. Fu, Rockburst prediction model based on comprehensive weight and extension methods and its engineering application, B Eng Geol Environ, 2020, 79(9): 4891–4903.

[24] D. Li, Z. Liu, D.J. Armaghani, P. Xiao, J. Zhou, Novel ensemble intelligence methodologies for rockburst assessment in complex and variable environments, Scientific Reports. 2022, 12(1): 1844.

[25] A. Rsf, A. At, C. Lresb, A. Mk, Rockburst assessment in deep geotechnical conditions using true-triaxial tests and data-driven approaches, Int J Rock Mech Min. 2020, 128: 104279.

Reviewer 4 Report (New Reviewer)

Author's have addressed comments according. Therefore, the paper is accepted for publication.

Author Response

to Reviewer’ Comments

Reviewer 4Author's have addressed comments according. Therefore, the paper is accepted for publication.

Reply: Thanks very much for your comments and suggestions.

This manuscript is a resubmission of an earlier submission. The following is a list of the peer review reports and author responses from that submission.

Round 1

Reviewer 1 Report

The topic is interesting however, it should be rewritten, and to add  subsections to the actual version

1. Please rewrite the title of the paper to be more clear for readers

2. several orthographic errors such as lines 29, 276,272 ...

3. line 32: I think this term is not appropriate in this case (Cancer)

4. Figure 1. please cite the related  reference of this statement and figure

5. line 42: This sentence should remove as there is no relation between the spatial distribution of the mines in 2D with rock bursts in this very specific study

6. line 56: The emphasis here is more on the rock burst in coal mines which contradicts other statements. 

7. The introduction is quite modest and doesn't cover the state of the art of the several criteria applied for the prediction of rock burst. The introduction should be rewritten, mentioning the other approaches, their pros, and cons, and why the new criteria are more useful,...

8. Table 1: there are many other non-energetic criteria that are not mentioned !!!

9. line 66: each index should explained briefly and mention why their calculation is more complex

10. line 69: this sentence should rewrite, it is not clear

11. line 92: these statements are not coherent, please revise them

12. line 95: Why? please explain the reason why this index is more suitable

13. line 97: may be better to replace it with by the authors of the present study

14. line 101: the limitations of these criteria should be mentioned. How do these criteria consider the role of discontinuities? how they can upscale?

15. the relation between these indexes and the mechanism of the rock burst should be explained

16. line 127: please cite the reference of this statement?!!

17. line 184: the  loading velocity test is necessary as every rock sample is unic

18. Table 4. in this case, how the authors can calculate and monitor the post-peak?

19. lines 201 and 202: what is the unit?

20. line 207: these observations should be explained clearly 

21. line 214: What is the novelty of this criteria? is well known the relationship between the loading rate and rock strength in rock mechanics and solid mechanics society

22. Figure 8. the observation of the figures is not consistent with the statement

23. line 338: the results of this criteria are almost the same as the criteria of the E before and after post-peck. what it is the novelty and advantages?

Reviewer 2 Report

1. The reference list is absolutely unbalanced - the reference cites only research made in China. All western studies are absolutely ignored.

2. The physical basis for the criteria is not clear. It is not clear which criteria were put forward by the authors and which were put forward in previous studies.

3. There is no confirmation of the correctness of the criteria boundaries.

4. English style must be significantly improved.

5. The abstract and conclusion are written in a very generic form and it is not clear what is the novelty of the research.

3. The limits for 

Reviewer 3 Report

minor revision
